# Taming Local Effects in Graph-based Spatiotemporal Forecasting

**Andrea Cini** [*1], **Ivan Marisca** [*1], **Daniele Zambon** [1], **Cesare Alippi** [12]

[1] The Swiss AI Lab IDSIA USI-SUPSI, Università della Svizzera italiana, [2] Politecnico di Milano

`{andrea.cini, ivan.marisca, daniele.zambon, cesare.alippi}@usi.ch`

## Abstract

Spatiotemporal graph neural networks have shown to be effective in time series forecasting applications, achieving better performance than standard univariate predictors in several settings. These architectures take advantage of a graph structure and relational inductive biases to learn a single (*global*) inductive model to predict any number of the input time series, each associated with a graph node. Despite the gain achieved in computational and data efficiency w.r.t. fitting a set of *local* models, relying on a single global model can be a limitation whenever some of the time series are generated by a different spatiotemporal stochastic process. The main objective of this paper is to understand the interplay between *globality* and *locality* in graph-based spatiotemporal forecasting, while contextually proposing a methodological framework to rationalize the practice of including trainable node embeddings in such architectures. We ascribe to trainable node embeddings the role of amortizing the learning of specialized components. Moreover, embeddings allow for 1) effectively combining the advantages of shared message-passing layers with node-specific parameters and 2) efficiently transferring the learned model to new node sets. Supported by strong empirical evidence, we provide insights and guidelines for specializing graph-based models to the dynamics of each time series and show how this aspect plays a crucial role in obtaining accurate predictions.

## 1 Introduction

Neural forecasting methods [1] take advantage of large databases of related time series to learn models for each individual process. If the time series in the database are not independent, functional dependencies among them can be exploited to obtain more accurate predictions, e.g., when the considered time series are observations collected from a network of sensors. In this setting, we use the term *spatiotemporal time series* to indicate the existence of relationships among subsets of time series (sensors) that span additional axes other than the temporal one, denoted here as *spatial* in a broad sense. Graph representations effectively model such dependencies and graph neural networks (GNNs) [2–4] can be included as modules in the forecasting architecture to propagate information along the spatial dimension. The resulting neural architectures are known in the literature as spatiotemporal graph neural networks (STGNNs) [5, 6] and have found widespread adoption in relevant applications ranging from traffic forecasting [6, 7] to energy analytics [8, 9]. These models embed inductive biases typical of graph deep learning and graph signal processing [10] and, thus, have several advantages over standard multivariate models; in fact, a shared set of learnable weights is used to obtain predictions for each time series by conditioning on observations at the neighboring nodes. Nonetheless, it has become more and more common to see node-specific trainable parameters being introduced as means to extract node(sensor)-level features then used as spatial identifiers within the processing [11, 12]. By doing so, the designer accepts a compromise in transferability that often

---

[*]Equal contribution.

empirically leads to higher forecasting accuracy on the task at hand. We argue that the community has yet to find a proper explanatory framework for this phenomenon and, notably, has yet to design proper methodologies to deal with the root causes of observed empirical results, in practical applications.

In the broader context of time series forecasting, single models learned from a set of time series are categorized as *global* and are opposed to *local* models, which instead are specialized for any particular time series in the set [13, 14]. Global models are usually more robust than local ones, as they require a smaller total number of parameters and are fitted on more samples. Standard STGNNs, then, fall within the class of global models and, as a result, often have an advantage over local multivariate approaches; however, explicitly accounting for the behavior of individual time series might be problematic and require a large memory and model capacity. As an example, consider the problem of electric load forecasting: consumption patterns of single customers are influenced by shared factors, e.g., weather conditions and holidays, but are also determined by the daily routine of the individual users related by varying degrees of affinity. We refer to the dynamics proper to individual nodes as *local effects*.

Local effects can be accounted for by combining a global model with local components, e.g., by using encoding and/or decoding layers specialized for each input time series paired with a core global processing block. While such an approach to building hybrid global-local models can be effective, the added model complexity and specialization can negate the benefits of using a global component. In this paper, we propose to consider and interpret learned node embeddings as a mechanism to amortize the learning of local components; in fact, instead of learning a separate processing layer for each time series, node embeddings allow for learning a single global module conditioned on the learned (local) node features. Furthermore, we show that – within a proper methodological framework – node embeddings can be fitted to different node sets, thus enabling an effective and efficient transfer of the core processing modules.

**Contributions** In this paper, we analyze the effect of node-specific components in spatiotemporal time series models and assess how to incorporate them in the forecasting architecture, while providing an understanding of each design choice within a proper context. The major findings can be summarized in the following statements.

**S1** Local components can be crucial to obtain accurate predictions in spatiotemporal forecasting.

**S2** Node embeddings can amortize the learning of local components.

**S3** Hybrid global-local STGNNs can capture local effects with contained model complexity and smaller input windows w.r.t. fully global approaches.

**S4** Node embeddings for time series outside the training dataset can be obtained by fitting a relatively small number of observations and yield more effective transferability than fine-tuning global models.

**S5** Giving structure to the embedding space provides an effective regularization, allowing for similarities among time series to emerge and shedding light on the role of local embeddings within a global architecture.

Throughout the paper, we reference the findings with their pointers. Against this backdrop, our main novel contributions reside in:

- A sound conceptual and methodological framework for dealing with local effects and designing node-specific components in STGNN architectures.

- An assessment of the role of learnable node embeddings in STGNNs and methods to obtain them.

- A comprehensive empirical analysis of the aforementioned phenomena in representative architectures across synthetic and real-world datasets.

- Methods to structure the embedding space, thus allowing for effective and efficient reuse of the global component in a transfer learning scenario.

We believe that our study constitutes an essential advancement toward the understanding of the interplay of different inductive biases in graph-based predictors, and argue that the methodologies and conceptual developments proposed in this work will constitute a foundational piece of know-how for the practitioner.

## 2  Preliminaries and problem statement

Consider a set of $N$ time series with graph-side information where the $i$-th time series is composed by a sequence of $d_x$ dimensional vectors $\boldsymbol{x}_t^i \in \mathbb{R}^{d_x}$ observed at time step $t$; each time series $\{\boldsymbol{x}_t^i\}_t$ might be generated by a different stochastic process. Matrix $\boldsymbol{X}_t \in \mathbb{R}^{N \times d_x}$ encompasses the $N$ observations at time $t$ and, similarly, $\boldsymbol{X}_{t:t+T}$ indicates the sequence of observations within the time interval $[t, t+T)$. Relational information is encoded by a weighted adjacency matrix $\boldsymbol{A} \in \mathbb{R}^{N \times N}$ that accounts for (soft) functional dependencies existing among the different time series. We use interchangeably the terms *node* and *sensor* to indicate the entities generating the time series and refer to the node set together with the relational information (graph) as *sensor network*. Eventual exogenous variables associated with each node are indicated by $\boldsymbol{U}_t \in \mathbb{R}^{N \times d_u}$; the tuple $\mathcal{G}_t = \langle \boldsymbol{X}_t, \boldsymbol{U}_t, \boldsymbol{A} \rangle$ indicates all the available information associated with time step $t$.

We address the multistep-ahead time-series forecasting problem, i.e., we are interested in predicting, for every time step $t$ and some $H, W \geq 1$, the expected value of the next $H$ observations $\boldsymbol{X}_{t:t+H}$ given a window $\mathcal{G}_{t-W:t}$ of $W$ past measurements. In case data from multiple sensor networks are available, the problem can be formalized as learning from $M$ disjoint collections of spatiotemporal time series $\mathcal{D} = \big\{ \mathcal{G}_{t_1:t_1+T_1}^{(1)}, \mathcal{G}_{t_2:t_2+T_2}^{(2)}, \dots, \mathcal{G}_{t_m:t_m+T_m}^{(M)} \big\}$, potentially without overlapping time frames. In the latter case, we assume sensors to be homogeneous both within a single network and among different sets. Furthermore, we assume edges to indicate the same type of relational dependencies, e.g., physical proximity.

## 3  Forecasting with Spatiotemporal Graph Neural Networks

This section provides a taxonomy of the different components that constitute an STGNN. Based on the resulting archetypes, reference operators for this study are identified. The last paragraph of the section broadens the analysis to fit STGNNs within more general time series forecasting frameworks.

**Spatiotemporal message-passing**  We consider STGNNs obtained by stacking spatiotemporal message-passing (STMP) layers s.t.

$$\boldsymbol{H}_t^{l+1} = \text{STMP}^l \left( \boldsymbol{H}_{\leq t}^l, \boldsymbol{A} \right), \tag{1}$$

where $\boldsymbol{H}_t^l \in \mathbb{R}^{N \times d_h}$ indicates the stack of node representations $\boldsymbol{h}_t^{i,l}$ at time step $t$ at the $l$-th layer. The shorthand $\leq t$ indicates the sequence of all representations corresponding to the time steps up to $t$ (included). Each $\text{STMP}^l(\,\cdot\,)$ layer is structured as follows

$$\boldsymbol{h}_t^{i,l+1} = \rho^l \Big( \boldsymbol{h}_{\leq t}^{i,l}, \underset{j \in \mathcal{N}(i)}{\text{AGGR}} \big\{ \gamma^l \big( \boldsymbol{h}_{\leq t}^{i,l}, \boldsymbol{h}_{\leq t}^{j,l}, a_{ji} \big) \big\} \Big), \tag{2}$$

where $\rho^l$ and $\gamma^l$ are respectively the update and message functions, e.g., implemented by multilayer perceptrons (MLPs) or recurrent neural networks (RNNs). $\text{AGGR}\{\cdot\}$ indicates a generic permutation invariant aggregation function, while $\mathcal{N}(i)$ refers to the set of neighbors of node $i$, each associated with an edge with weight $a_{ji}$. Models of this type are fully *inductive*, in the sense that they can be used to make predictions for networks and time windows different from those they have been trained on, provided a certain level of similarity (e.g., homogenous sensors) between source and target node sets [15].

Among the different implementations of this general framework, we can distinguish between time-then-space (TTS) and time-and-space (T&S) models by following the terminology of previous works [16, 17]. Specifically, in TTS models the sequence of representations $\boldsymbol{h}_{\leq t}^{i,0}$ is encoded by a sequence model, e.g., an RNN, before propagating information along the spatial dimension through message passing (MP) [16]. Conversely, in T&S models time and space are processed in a more integrated fashion, e.g., by a recurrent GNN [5] or by spatiotemporal convolutional operators [7]. In the remainder of the paper, we take for TTS model an STGNN composed by a Gated Recurrent Unit (GRU) [18] followed by standard MP layers [19]:

$$\boldsymbol{H}_t^1 = \text{GRU}\left( \boldsymbol{H}_{\leq t}^0 \right), \qquad\qquad \boldsymbol{H}_t^{l+1} = \text{MP}^l \left( \boldsymbol{H}_t^l, \boldsymbol{A} \right), \tag{3}$$

where $l = 1, \dots, L-1$ and $\text{GRU}(\,\cdot\,)$ processes sequences node-wise. Similarly, we consider as reference T&S model a GRU with an MP network at its gates [5, 20], that process input data as

$$\boldsymbol{H}_t^{l+1} = \text{GRU}^l \left( \big\{ \text{MP}^l \left( \boldsymbol{H}_t^l, \boldsymbol{A} \right) \big\}_{\leq t} \right). \tag{4}$$

Moreover, STGNN models can be further categorized w.r.t. the implementation of message function; in particular, by loosely following Dwivedi et al. [21], we call *isotropic* those GNNs where the message function $\gamma^l$ only depends on the features of the sender node $\boldsymbol{h}_{\leq t}^{j,l}$; conversely, we call *anistropic* GNNs where $\gamma^l$ takes both $\boldsymbol{h}_{\leq t}^{i,l}$ and $\boldsymbol{h}_{\leq t}^{j,l}$ as input. In the following, the case-study isotropic operator is

$$\boldsymbol{h}_t^{i,l+1} = \xi\left(\boldsymbol{W}_1^l \boldsymbol{h}_t^{i,l} + \underset{j \in \mathcal{N}(i)}{\text{MEAN}}\left\{\boldsymbol{W}_2^l \boldsymbol{h}_t^{j,l}\right\}\right), \tag{5}$$

where $\boldsymbol{W}_1^l$ and $\boldsymbol{W}_2^l$ are matrices of learnable parameters and $\xi(\cdot)$ a generic activation function. Conversely, the operator of choice for the anisotropic case corresponds to

$$\boldsymbol{m}_t^{j \to i,l} = \boldsymbol{W}_2^l \xi\left(\boldsymbol{W}_1^l\left[\boldsymbol{h}_t^{i,l}||\boldsymbol{h}_t^{j,l}||a_{ji}\right]\right), \qquad \alpha_t^{j \to i,l} = \sigma\left(\boldsymbol{W}_0^l \boldsymbol{m}_t^{j \to i,l}\right), \tag{6}$$

$$\boldsymbol{h}_t^{i,l+1} = \xi\left(\boldsymbol{W}_3^l \boldsymbol{h}_t^{i,l} + \underset{j \in \mathcal{N}(i)}{\text{SUM}}\left\{\alpha_t^{j \to i,l} \boldsymbol{m}_t^{j \to i,l}\right\}\right), \tag{7}$$

where matrices $\boldsymbol{W}_0^l \in \mathbb{R}^{1 \times d_m}$, $\boldsymbol{W}_1^l$, $\boldsymbol{W}_2^l$ and $\boldsymbol{W}_3^l$ are learnable parameters, $\sigma(\cdot)$ is the sigmoid activation function and $||$ the concatenation operator applied along the feature dimension (see Appendix A.2 for a detailed description).

**Global and local forecasting models**  Formally, a time series forecasting model is called *global* if its parameters are fitted to a group of time series (either univariate or multivariate), while *local* models are specific to a single (possibly multivariate) time series. The advantages of global models have been discussed at length in the time series forecasting literature [22, 14, 13, 1] and are mainly ascribable to the availability of large amounts of data that enable generalization and the use of models with higher capacity w.r.t. the single local models. As presented, and further detailed in the following section, STGNNs are global, yet have a peculiar position in this context, as they exploit spatial dependencies localizing predictions w.r.t. each node's neighborhood. Furthermore, the transferability of GNNs makes these models distinctively different from local multivariate approaches enabling their use in cold-start scenarios [23] and making them inductive both temporally and spatially.

## 4  Locality and globality in Spatiotemporal Graph Neural Networks

We now focus on the impact of local effects in forecasting architectures based on STGNNs. The section starts by introducing a template to combine different processing layers within a global model, then continues by discussing how these can be turned into local. Contextually, we start to empirically probe the reference architectures.

### 4.1  A global processing template

STGNNs localize predictions in space, i.e., with respect to a single node, by exploiting an MP operator that contextualizes predictions by constraining the information flow within each node's neighborhood. STGNNs are global forecasting models s.t.

$$\widehat{\boldsymbol{X}}_{t:t+H} = F_{\text{G}}\left(\mathcal{G}_{t-W:t}; \phi\right) \tag{8}$$

where $\phi$ are the learnable parameters shared among the time series and $\widehat{\boldsymbol{X}}_{t:t+H}$ indicate the $H$-step ahead predictions of the input time series given the window of (structured) observations $\boldsymbol{X}_{t-W:t}$. In particular, we consider forecasting architectures consisting of an encoding step followed by STMP layers and a final readout mapping representations to predictions; the corresponding sequence of operations composing $F_{\text{G}}$ can be summarized as

$$\boldsymbol{h}_t^{i,0} = \text{ENCODER}\left(\boldsymbol{x}_{t-1}^i, \boldsymbol{u}_{t-1}^i\right), \tag{9}$$

$$\boldsymbol{H}_t^{l+1} = \text{STMP}^l\left(\boldsymbol{H}_{\leq t}^l, \boldsymbol{A}\right), \quad l = 0, \dots, L-1 \tag{10}$$

$$\hat{\boldsymbol{x}}_{t:t+H}^i = \text{DECODER}\left(\boldsymbol{h}_t^{i,L}\right). \tag{11}$$

ENCODER($\cdot$) and DECODER($\cdot$) indicate generic encoder and readout layers that could be implemented in several ways. In the following, the encoder is assumed to be a standard fully connected linear layer, while the decoder is implemented by an MLP with a single hidden layer followed by an output (linear) layer for each forecasting step.

## 4.2 Local effects

Differently from the global approach, local models are fitted to a single time series (e.g., see the standard Box-Jenkins approach) and, in our problem settings, can be indicated as

$$\hat{\boldsymbol{x}}_{t:t+H}^i = f_i\left(\boldsymbol{x}_{t-W:t}^i; \theta^i\right), \tag{12}$$

where $f_i\left(\,\cdot\,; \theta^i\right)$ is a sensor-specific model, e.g., an RNN with its dedicated parameters $\theta^i$, fitted on the $i$-th time series. While the advantages of global models have already been discussed, local effects, i.e., the dynamics observed at the level of the single sensor, are potentially more easily captured by a local model. In fact, if local effects are present, global models might require an impractically large model capacity to account for all node-specific dynamics [13], thus losing some of the advantages of using a global approach (**S3**). In the STGNN case, then, increasing the input window for each node would result in a large computational overhead. Conversely, purely local approaches fail to exploit relational information among the time series and cannot reuse available knowledge efficiently in an inductive setting.

Combining global graph-based components with local node-level components has the potential for achieving a two-fold objective: 1) exploiting relational dependencies together with side information to learn flexible and efficient graph deep learning models and 2) making at the same time specialized and accurate predictions for each time series. In particular, we indicate global-local STGNNs as

$$\hat{\boldsymbol{x}}_{t:t+H}^i = F\left(\mathcal{G}_{t-W:t}; \phi, \theta^i\right) \tag{13}$$

where function $F$ and parameter vector $\phi$ are shared across all nodes, whereas parameter vector $\theta^i$ is time-series dependent. Such a function $F(\,\cdot\,)$ could be implemented, for example, as a sum between a global model (Eq. 8) and a local one (Eq. 12):

$$\widehat{\boldsymbol{X}}_{t:t+H}^{(1)} = F_{\mathrm{G}}\left(\mathcal{G}_{t-W:t}; \phi\right), \qquad \hat{\boldsymbol{x}}_{t:t+H}^{i,(2)} = f_i\left(\boldsymbol{x}_{t-W:t}^i; \theta^i\right), \tag{14}$$

$$\hat{\boldsymbol{x}}_{t:t+H}^i = \hat{\boldsymbol{x}}_{t:t+H}^{i,(1)} + \hat{\boldsymbol{x}}_{t:t+H}^{i,(2)}, \tag{15}$$

or – with a more integrated approach – by using different weights for each time series at the encoding and/or decoding steps. The latter approach results in using a different encoder and/or decoder for each $i$-th node in the template STGNN (Eq. 9–11) to extract representations and, eventually, project them back into input space:

$$\boldsymbol{h}_t^{i,0} = \text{ENCODER}_i\left(\boldsymbol{x}_{t-1}^i, \boldsymbol{u}_{t-1}^i; \theta_{enc}^i\right), \quad (16) \qquad \hat{\boldsymbol{x}}_{t:t+H}^i = \text{DECODER}_i\left(\boldsymbol{h}_t^{i,L}; \theta_{dec}^i\right). \tag{17}$$

MP layers could in principle be specialized as well, e.g., by using a different local update function $\gamma_i(\,\cdot\,)$ for each node. However, this would be impractical unless subsets of nodes are allowed to share parameters to some extent (e.g., by clustering them).

To support our arguments, Tab. 1 shows empirical results for the reference TTS models with isotropic message passing (TTS-IMP) on 2 popular traffic forecasting benchmarks (METR-LA and PEMS-BAY [6]). In particular, we compare the global approach with 3 hybrid global-local variants where local weights are used in the encoder, in the decoder, or in both of them (see Eq. 16-17 and the light brown block in Tab. 1). Notably, while fitting a separate RNN to each individual time series fails (LocalRNNs), exploiting a local encoder and/or decoder significantly improves performance w.r.t. the fully global model (**S1**). Note that the price of specialization is paid in terms of the number of learnable parameters which is an order of magnitude higher in global-local

Table 1: Perfomance (MAE) of TTS-IMP variants and number of associated trainable parameters in PEMS-BAY (5-run average).

| | | | **METR-LA** | **PEMS-BAY** | (# weights) |
|---|---|---|---|---|---|
| | | Global TTS | $3.35 \pm 0.01$ | $1.72 \pm 0.00$ | $4.71 \times 10^4$ |
| Global-local TTS | Weights | Encoder | $3.15 \pm 0.01$ | $1.66 \pm 0.01$ | $2.75 \times 10^5$ |
| | | Decoder | $3.09 \pm 0.01$ | $1.58 \pm 0.00$ | $3.00 \times 10^5$ |
| | | Enc. + Dec. | $3.16 \pm 0.01$ | $1.70 \pm 0.01$ | $5.28 \times 10^5$ |
| | Embed. | Encoder | $3.08 \pm 0.01$ | $1.58 \pm 0.00$ | $5.96 \times 10^4$ |
| | | Decoder | $3.13 \pm 0.00$ | $1.60 \pm 0.00$ | $5.96 \times 10^4$ |
| | | Enc. + Dec. | $3.07 \pm 0.01$ | $1.58 \pm 0.00$ | $6.16 \times 10^4$ |
| | | FC-RNN | $3.56 \pm 0.03$ | $2.32 \pm 0.01$ | $3.04 \times 10^5$ |
| | | LocalRNNs | $3.69 \pm 0.00$ | $1.91 \pm 0.00$ | $1.10 \times 10^7$ |

variants. The table reports as a reference also results for FC-RNN, a multivariate RNN taking as input the concatenation of all time series. Indeed, having both encoder and decoder implemented as local layers leads to a large number of parameters and has a marginal impact on forecasting accuracy. The

light gray block in Tab. 1 anticipates the effect of replacing the local layers with the use of learnable *node embeddings*, an approach discussed in depth in the next section. Additional results including TTS models with anisotropic message passing (TTS-AMP) are provided in Appendix B.

## 5 Node embeddings

This section introduces node embeddings as a mechanism to amortize the learning of local components and discusses the supporting empirical results. We then propose possible regularization techniques and discuss the advantages of embeddings in transfer learning scenarios.

**Amortized specialization**  Static node features offer the opportunity to design and obtain node identification mechanisms across different time windows to tailor predictions to a specific node. However, in most settings, node features are either unavailable or insufficient to characterize the node dynamics. A possible solution consists of resorting to learnable node embeddings, i.e., a table of learnable parameters $\Theta = \boldsymbol{V} \in \mathbb{R}^{N \times d_v}$. Rather than interpret these learned representations as positional encodings, our proposal is to consider them as a way of amortizing the learning of node-level specialized models. More specifically, instead of learning a local model for each time series, embeddings fed into modules of a global STGNN and learned end-to-end with the forecasting architecture allow for specializing predictions by simply relying on gradient descent to find a suitable encoding.

The most straightforward option for feeding embeddings into the processing is to update the template model by changing the encoder and decoder as

$$\boldsymbol{h}_t^{i,0} = \text{ENCODER}\left(\boldsymbol{x}_{t-1}^i, \boldsymbol{u}_{t-1}^i, \boldsymbol{v}^i\right), \quad (18) \qquad \hat{\boldsymbol{x}}_{t:t+H}^i = \text{DECODER}\left(\boldsymbol{h}_t^{i,L}, \boldsymbol{v}^i\right). \quad (19)$$

which can be seen as amortized versions of the encoder and decoder in Eq. 16-17. The encoding scheme of Eq. 18 also facilitates the propagation of relevant information by identifying nodes, an aspect that can be particularly significant as message-passing operators – in particular isotropic ones – can act as low-pass filters that smooth out node-level features [24, 25].

Tab. 1 (light gray block) reports empirical results that show the effectiveness of embeddings in amortizing the learning of local components (**S2**), with a negligible increase in the number of trainable parameters w.r.t. the base global model. In particular, feeding embeddings to the encoder, instead of conditioning the decoding step only, results in markedly better performance, hinting at the impact of providing node identification ahead of MP (additional empirical results provided in Sec. 7).

### 5.1 Structuring the embedding space

The latent space in which embeddings are learned can be structured and regularized to gather benefits in terms of interpretability, transferability, and generality (**S5**). In fact, accommodating new embeddings can be problematic, as they must fit in a region of the embedding space where the trained model can operate, and, at the same time, capture the local effects at the new nodes. In this setting, proper regularization can provide guidance and positive inductive biases to transfer the learned model to different node sets. As an example, if domain knowledge suggests that neighboring nodes have similar dynamics, Laplacian regularization [26, 27] can be added to the loss. In the following, we propose two general strategies based respectively on variational inference and on a clustering loss to impose soft constraints on the latent space. As shown in Sec. 7, the resulting structured space additionally allows us to gather insights into the features encoded in the embeddings.

**Variational regularization**  As a probabilistic approach to structuring the latent space, we propose to consider learned embeddings as parameters of an approximate posterior distribution – given the training data – on the vector used to condition the predictions. In practice, we model each node embedding as a sample from a multivariate Gaussian $\boldsymbol{v}^i \sim q^i(\boldsymbol{v}^i|\mathcal{D}) = \mathcal{N}\left(\boldsymbol{\mu}_i, \text{diag}(\boldsymbol{\sigma}_i^2)\right)$ where $(\boldsymbol{\mu}_i, \boldsymbol{\sigma}_i)$ are the learnable (local) parameters. Each node-level distribution is fitted on the training data by considering a standard Gaussian prior and exploiting the reparametrization trick [28] to minimize under the sampling of $t$

$$\delta_t \doteq \mathbb{E}_{\boldsymbol{V} \sim Q}\left[\left\|\widehat{\boldsymbol{X}}_{t:t+H} - \boldsymbol{X}_{t:t+H}\right\|_2^2\right] + \beta D_{\text{KL}}(Q|P),$$

where $P = \mathcal{N}(\mathbf{0}, \mathbb{I})$ is the prior, $D_{\mathrm{KL}}$ the Kulback-Leibler divergence, and $\beta$ controls the regularization strength. This regularization scheme results in a smooth latent space where it is easier to interpolate between representations, thus providing a principled way for accommodating different node embeddings.

**Clustering regularization**   A different (and potentially complementary) approach to structuring the latent space is to incentivize node embeddings to form clusters and, consequently, to self-organize into different groups. We do so by introducing a regularization loss inspired by deep $K$-means algorithms [29]. In particular, besides the embedding table $\boldsymbol{V} \in \mathbb{R}^{N \times d_v}$, we equip the embedding module with a matrix $\boldsymbol{C} \in \mathbb{R}^{K \times d_v}$ of $K \ll N$ learnable centroids and a cluster assignment matrix $\boldsymbol{S} \in \mathbb{R}^{N \times K}$ encoding scores associated to each node-cluster pair. We consider scores as logits of a categorical (Boltzmann) distribution and learn them by minimizing the regularization term

$$\mathcal{L}_{reg} \doteq \mathbb{E}_{\boldsymbol{M}}\left[\|\boldsymbol{V} - \boldsymbol{M}\boldsymbol{C}\|_2\right], \quad p(\boldsymbol{M}_{ij} = 1) = \frac{e^{\boldsymbol{S}_{ij}/\tau}}{\sum e^{\boldsymbol{S}_{ik}/\tau}},$$

where $\tau$ is a hyperparameter. We minimize $\mathcal{L}_{reg}$ – which corresponds to the embedding-to-centroid distance – jointly with the forecasting loss by relying on the Gumbel softmax trick [30]. Similarly to the variational inference approach, the clustering regularization gives structure to embedding space and allows for inspecting patterns in the learned local components (see Sec. 7).

**Transferability of graph-based predictors**   Global models based on GNNs can make predictions for never-seen-before node sets, and handle graphs of different sizes and variable topology. In practice, this means that the graph-based predictors can easily handle new sensors being added to the network over time and be used for zero-shot transfer. Clearly, including in the forecasting architecture node-specific local components compromises these properties. Luckily, if local components are replaced by node embedding, adapting the specialized components is relatively cheap since the number of parameters to fit w.r.t. the new context is usually contained, and – eventually – both the graph topology and the structure of the embedding latent space can be exploited (**S4**). Experiments in Sec. 7 provide an in-depth empirical analysis of transferability within our framework and show that the discussed regularizations can be useful in this regard.

## 6   Related works

GNNs have been remarkably successful in modeling structured dynamical systems [31–33], temporal networks [34–36] and sequences of graphs [37, 38]. For what concerns time series processing, recurrent GNNs [5, 6] were among the first STGNNs being developed, followed by fully convolutional models [7, 39] and attention-based solutions [40–42]. Among the methods that focus on modeling node-specific dynamics, Bai et al. [11] use a factorization of the weight matrices in a recurrent STGNN to adapt the extracted representation to each node. Conversely, Chen et al. [43] use a model inspired by Wang et al. [44] consisting of a global GNN paired with a local model conditioned on the neighborhood of each node. Node embeddings have been mainly used in structure-learning modules to amortize the cost of learning the full adjacency matrix [39, 45, 12] and in attention-based approaches as positional encodings [46, 47, 41]. Shao et al. [48] observed how adding spatiotemporal identification mechanisms to the forecasting architecture can outperform several state-of-the-art STGNNs. Conversely, Yin et al. [49] used a cluster-based regularization to fine-tune an AGCRN-like model on different datasets. However, none of the previous works systematically addressed directly the problem of globality and locality in STGNNs, nor provided a comprehensive framework accounting for learnable node embeddings within different settings and architectures. Finally, besides STGNNs, there are several examples of hybrid global and local time series forecasting models. Wang et al. [44] propose an architecture where $K$ global models extract dynamic global factors that are then weighted and integrated with probabilistic local models. Sen et al. [50] instead use a matrix factorization scheme paired with a temporal convolutional network [51] to learn a multivariate model then used to condition a second local predictor.

## 7   Experiments

This section reports salient results of an extensive empirical analysis of global and local models and combinations thereof in spatiotemporal forecasting benchmarks and different problem settings;

complete results of this systematic analysis can be found in Appendix B. Besides the reference architectures, we consider the following baselines and popular state-of-the-art architectures.

**RNN:** a global univariate RNN sharing the same parameters across the time series.

**FC-RNN:** a multivariate RNN taking as input the time series as if they were a multivariate one.

**LocalRNNs:** local univariate RNNs with different sets of parameters for each time series.

**DCRNN [6]:** a recurrent T&S model with the Diffusion Convolutional operator.

**AGCRN [11]:** the T&S global-local Adaptive Graph Convolutional Recurrent Network.

**GraphWaveNet:** the deep T&S spatiotemporal convolutional network by Wu et al. [39].

We also consider a global-local RNN, in which we specialize the model by using node embeddings in the encoder. Note that among the methods selected from the literature only DCRNN can be considered fully global (see Sec. 6). Performance is measured in terms of *mean absolute error* (MAE).

**Synthetic data** We start by assessing the performance of hybrid global-local spatiotemporal models in a controlled environment, considering a variation of GP-VAR [52], a synthetic dataset based on a polynomial graph filter [53], that we modify to include local effects. In particular, data are generated from the spatiotemporal process

$$\boldsymbol{H}_t = \sum_{l=1}^{L} \sum_{q=1}^{Q} \Theta_{q,l} \boldsymbol{A}^{l-1} \boldsymbol{X}_{t-q},$$

$$\boldsymbol{X}_{t+1} = \boldsymbol{a} \odot \tanh\left(\boldsymbol{H}_t\right) + \boldsymbol{b} \odot \tanh\left(\boldsymbol{X}_{t-1}\right) + \eta_t, \quad (20)$$

where $\Theta \in \mathbb{R}^{Q \times L}$, $\boldsymbol{a} \in \mathbb{R}^N$, $\boldsymbol{b} \in \mathbb{R}^N$ and $\eta_t \sim \mathcal{N}(\boldsymbol{0}, \sigma^2 \mathbb{I})$. We refer to this dataset as GPVAR-L: note that $\boldsymbol{a}$ and $\boldsymbol{b}$ are node-specific parameters that inject local effects into the spatiotemporal process. We indicate simply as GPVAR the process obtained by fixing $\boldsymbol{a} = \boldsymbol{b} = \boldsymbol{0.5}$, i.e., by removing local effects. We use as the adjacency matrix the community graph used in prior works, increasing the number of nodes to 120 (see Appendix A.1).

Table 2: One-step-ahead forecasting error (MAE) of the different models in GP-VAR datasets (5 runs).

| | MODELS | GPVAR | GPVAR-L |
|---|---|---|---|
| | FC-RNN | .4393±.0024 | .5978±.0149 |
| | LocalRNNs | .4047±.0001 | .4610±.0003 |
| Global | RNN | .3999±.0000 | .5440±.0003 |
| Global | TTS-IMP | .3232±.0002 | .4059±.0032 |
| Global | TTS-AMP | .3193±.0000 | .3587±.0049 |
| Emb. | RNN | .3991±.0001 | .4612±.0003 |
| Emb. | TTS-IMP | .3195±.0000 | .3200±.0002 |
| Emb. | TTS-AMP | .3194±.0001 | .3199±.0001 |
| | Optimal model | .3192 | .3192 |

Tab. 2 shows forecasting accuracy for reference architectures with a 6-steps window on data generated from the processes. In the setting with no local effects, all STGNNs achieve performance close to the theoretical optimum, outperforming global and local univariate models but also the multivariate FC-RNN that – without any inductive bias – struggles to properly fit the data. In GPVAR-L, global and univariate models fail to match the performance of STGNNs that include local components (**S1**); interestingly, the global model with anisotropic MP outperforms the isotropic alternative, suggesting that the more advanced MP schemes can lead to more effective state identification.

Table 3: TTS-IMP one-step-ahead MAE on GPVAR-L with varying window length $W$ and capacity $d_h$ (5 runs).

| | Global | | | Embeddings | | |
|---|---|---|---|---|---|---|
| $W$ | $d_h = 16$ | $d_h = 32$ | $d_h = 64$ | $d_h = 16$ | $d_h = 32$ | $d_h = 64$ |
| 2 | .5371±.0014 | .4679±.0016 | .4124±.0021 | .3198±.0001 | .3199±.0001 | .3203±.0001 |
| 6 | .4059±.0032 | .3578±.0031 | .3365±.0006 | .3200±.0002 | .3201±.0001 | .3209±.0002 |
| 12 | .3672±.0035 | .3362±.0012 | .3280±.0003 | .3200±.0001 | .3200±.0000 | .3211±.0003 |
| 24 | .3485±.0032 | .3286±.0005 | .3250±.0001 | .3200±.0002 | .3200±.0000 | .3211±.0003 |

Finally, Tab. 3 shows that to match the performance of models with local components, in fully global models both window size and capacity should be increased (**S3**). This result is in line with what we should expect by considering the theory of local and global models and suggests that similar trade-offs might also happen in practical applications where both computational and sample complexity are a concern.

**Benchmarks** We then compare the performance of reference architectures and baselines with and without node embeddings at the encoding and decoding steps. Note that, while reference architectures and DCRNN are purely global models, GraphWaveNet and AGCRN use node embeddings to obtain an adjacency matrix for MP. AGCRN, furthermore, uses embeddings to make the convolutional filters

Table 4: Forecasting error (MAE) on 4 benchmark datasets (5 runs). The best result between each model and its variant with embeddings is in **bold**. N/A indicates runs exceeding resource capacity.

| MODELS | METR-LA | PEMS-BAY | CER-E | AQI | METR-LA | PEMS-BAY | CER-E | AQI |
|---|---|---|---|---|---|---|---|---|
| Reference arch. | Global models | | | | Global-local models (with embeddings) | | | |
| RNN | $3.54_{\pm.00}$ | $1.77_{\pm.00}$ | $456.98_{\pm0.61}$ | $14.02_{\pm.04}$ | $\mathbf{3.15}_{\pm.03}$ | $\mathbf{1.59}_{\pm.00}$ | $\mathbf{421.50}_{\pm1.78}$ | $\mathbf{13.73}_{\pm.04}$ |
| T&S-IMP | $3.35_{\pm.01}$ | $1.70_{\pm.01}$ | $443.85_{\pm0.99}$ | $12.87_{\pm.02}$ | $\mathbf{3.10}_{\pm.01}$ | $\mathbf{1.59}_{\pm.00}$ | $\mathbf{417.71}_{\pm1.28}$ | $\mathbf{12.48}_{\pm.03}$ |
| TTS-IMP | $3.34_{\pm.01}$ | $1.72_{\pm.00}$ | $439.13_{\pm0.51}$ | $12.74_{\pm.02}$ | $\mathbf{3.08}_{\pm.01}$ | $\mathbf{1.58}_{\pm.00}$ | $\mathbf{412.44}_{\pm2.80}$ | $\mathbf{12.33}_{\pm.02}$ |
| T&S-AMP | $3.22_{\pm.02}$ | $1.65_{\pm.00}$ | N/A | N/A | $\mathbf{3.07}_{\pm.02}$ | $1.59_{\pm.00}$ | N/A | N/A |
| TTS-AMP | $3.24_{\pm.01}$ | $1.66_{\pm.00}$ | $431.33_{\pm0.68}$ | $12.30_{\pm.02}$ | $\mathbf{3.06}_{\pm.01}$ | $\mathbf{1.58}_{\pm.01}$ | $\mathbf{412.95}_{\pm1.28}$ | $\mathbf{12.15}_{\pm.02}$ |
| Baseline arch. | Original | | | | Embeddings at Encoder and Decoder | | | |
| DCRNN | $3.22_{\pm.01}$ | $1.64_{\pm.00}$ | $428.36_{\pm1.23}$ | $12.96_{\pm.03}$ | $\mathbf{3.07}_{\pm.02}$ | $1.60_{\pm.00}$ | $\mathbf{412.87}_{\pm1.51}$ | $\mathbf{12.53}_{\pm.02}$ |
| GraphWaveNet | $3.05_{\pm.03}$ | $\mathbf{1.56}_{\pm.01}$ | $\mathbf{397.17}_{\pm0.67}$ | $12.08_{\pm.11}$ | $\mathbf{2.99}_{\pm.02}$ | $1.58_{\pm.00}$ | $401.15_{\pm1.49}$ | $\mathbf{11.81}_{\pm.04}$ |
| AGCRN | $3.16_{\pm.01}$ | $\mathbf{1.61}_{\pm.00}$ | $444.80_{\pm1.25}$ | $13.33_{\pm.02}$ | $\mathbf{3.14}_{\pm.00}$ | $1.62_{\pm.00}$ | $\mathbf{436.84}_{\pm2.06}$ | $\mathbf{13.28}_{\pm.03}$ |

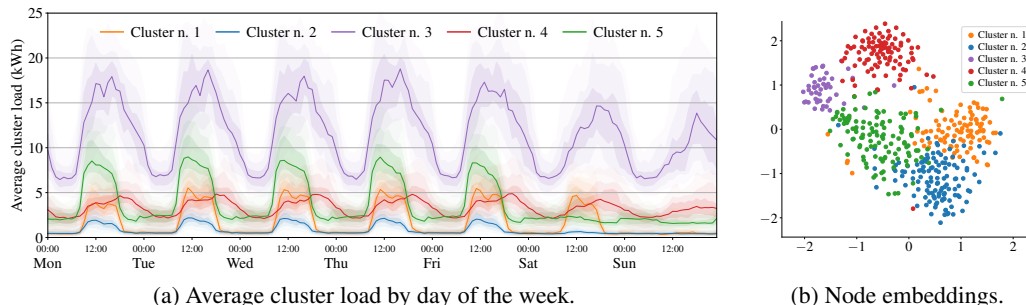

(a) Average cluster load by day of the week.      (b) Node embeddings.

Figure 1: Time series clusters in CER-E, obtained by regularizing the embedding space. **(a)** Average load for each clusters. **(b)** t-SNE plot of the corresponding node embeddings.

adaptive w.r.t. the node being processed. We evaluate all models on real-world datasets from three different domains: traffic networks, energy analytics, and air quality monitoring. Besides the already introduced traffic forecasting benchmarks (**METR-LA** and **PEMS-BAY**), we run experiments on smart metering data from the **CER-E** dataset [54] and air quality measurements from **AQI** [55] by using the same pre-processing steps and data splits of previous works [20, 47]. The full experimental setup is reported in appendix A. Tab. 4 reports forecasting *mean absolute error* (MAE) averaged over the forecasting horizon. Global-local reference models outperform the fully global variants in every considered scenario (**S1**). A similar observation can be made for the state-of-art architectures, where the impact of node embeddings (at encoding and decoding) is large for the fully global DCRNN and more contained in models already equipped with local components. Note that hyperparameters were not tuned to account for the change in architecture. Surprisingly, the simple TTS-IMP model equipped with node embeddings achieves results comparable to that of state-of-the-art STGNNs with a significantly lower number of parameters and a streamlined architecture. Interestingly, while both global and local RNNs models fail, the hybrid global-local RNN ranks among the best-performing models, outperforming graph-based models without node embeddings in most settings.

**Structured embeddings**      To test the hypothesis that structure in embedding space provides insights on the local effects at play (**S5**), we consider the clustering regularization method (Sec. 5.1) and the reference TTS-IMP model trained on the CER-E dataset. We set the number of learned centroids to $K = 5$ and train the cluster assignment mechanism end-to-end with the forecasting architecture. Then, we inspect the clustering assignment by looking at intra-cluster statistics. In particular, for each load profile, we compute the weekly average load curve, and, for each hour, we look at quantiles of the energy consumption within each cluster. Fig. 1a shows the results of the analysis by reporting the *median* load profile for each cluster; shaded areas correspond to quantiles with $10\%$ increments. Results show that users in the different clusters have distinctly different consumption patterns. Fig. 1b

Table 5: Forecasting error (MAE) in the transfer learning setting (5 runs average). Results refer to a 1-week fine-tuning set size on all PEMS datasets.

| | TTS-IMP | PEMS03 | PEMS04 | PEMS07 | PEMS08 |
|---|---|---|---|---|---|
| Fine-tuning | Global | $15.30 \pm 0.03$ | $21.59 \pm 0.11$ | $23.82 \pm 0.03$ | $15.90 \pm 0.07$ |
| | Embeddings | $14.64 \pm 0.05$ | $20.27 \pm 0.11$ | $\mathbf{22.23} \pm \mathbf{0.08}$ | $\mathbf{15.45} \pm \mathbf{0.06}$ |
| | – Variational | $\mathbf{14.56} \pm \mathbf{0.03}$ | $20.19 \pm 0.05$ | $22.43 \pm 0.02$ | $\mathbf{15.41} \pm \mathbf{0.06}$ |
| | – Clustering | $\mathbf{14.60} \pm \mathbf{0.02}$ | $\mathbf{19.91} \pm \mathbf{0.11}$ | $22.16 \pm 0.07$ | $\mathbf{15.41} \pm \mathbf{0.06}$ |
| | Zero-shot | $18.20 \pm 0.09$ | $23.88 \pm 0.08$ | $32.76 \pm 0.69$ | $20.41 \pm 0.07$ |

Table 6: Forecasting error (MAE) on PEMS04 in the transfer learning setting by varying fine-tuning set size (5 runs average).

| Model | Training set size | | | |
|---|---|---|---|---|
| TTS-IMP | 2 weeks | 1 week | 3 days | 1 day |
| Global | $20.86 \pm 0.03$ | $21.59 \pm 0.11$ | $21.84 \pm 0.06$ | $22.26 \pm 0.10$ |
| Embeddings | $19.96 \pm 0.08$ | $20.27 \pm 0.11$ | $21.03 \pm 0.14$ | $21.99 \pm 0.13$ |
| – Variational | $19.94 \pm 0.08$ | $20.19 \pm 0.05$ | $20.71 \pm 0.12$ | $\mathbf{21.20} \pm \mathbf{0.15}$ |
| – Clustering | $\mathbf{19.69} \pm \mathbf{0.06}$ | $\mathbf{19.91} \pm \mathbf{0.11}$ | $\mathbf{20.48} \pm \mathbf{0.09}$ | $21.91 \pm 0.21$ |

shows a 2D t-SNE visualization of the learned node embeddings, providing a view of the latent space and the effects of the cluster-based regularization.

**Transfer** In this experiment, we consider the scenario in which an STGNN for traffic forecasting is trained by using data from multiple traffic networks and then used to make predictions for a disjoint set of sensors sampled from the same region. We use the **PEMS03**, **PEMS04**, **PEMS07**, and **PEMS08** datasets [56], which contain measurements from 4 different districts in California. We train models on 3 of the datasets, fine-tune on 1 week of data from the target left-out dataset, validate on the following week, and test on the week thereafter. We compare variants of TTS-IMP with and without embeddings fed into encoder and decoder. Together with the unconstrained embeddings, we also consider the variational and clustering regularization approaches introduced in Sec. 5.1. At the fine-tuning stage, the global model updates all of its parameters, while in the hybrid global-local approaches only the embeddings are fitted to the new data. Tab. 5 reports results for the described scenario. The fully global approach is outperformed by the hybrid architectures in all target datasets (**S4**). Besides the significant improvement in performance, adjusting only node embeddings retains performance on the source datasets. Furthermore, results show the positive effects of regularizing the embedding space in the transfer setting (**S5**). This is further confirmed by results in Tab. 6, which report, for PEMS04, how forecasting error changes in relation to the length of the fine-tuning window. We refer to Appendix B for an in-depth analysis of several additional transfer learning scenarios.

## 8 Conclusions

We investigate the impact of locality and globality in graph-based spatiotemporal forecasting architectures. We propose a framework to explain empirical results associated with the use of trainable node embeddings and discuss different architectures and regularization techniques to account for local effects. The proposed methodologies are thoroughly empirically validated and, although not inductive, prove to be effective in a transfer learning context. We argue that our work provides necessary and key methodologies for the understanding and design of effective graph-based spatiotemporal forecasting architectures. Future works can build on the results presented here and study alternative, and even more transferable, methods to account for local effects.

### Acknowledgements

This research was partly funded by the Swiss National Science Foundation under grant 204061: *High-Order Relations and Dynamics in Graph Neural Networks*.

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

# Appendix

The following sections provide additional details on the experimental setting and computational platform used to gather the results presented in the paper. Additionally, we also include supplementary empirical results and analysis.

## A  Experimental setup

Experimental setup and baselines have been developed with Python [57] by relying on the following open-source libraries:

- PyTorch [58];
- PyTorch Lightning [59];
- PyTorch Geometric [60]
- Torch Spatiotemporal [61];
- numpy [62];
- scikit-learn [63].

Experiments were run on a workstation equipped with AMD EPYC 7513 processors and four NVIDIA RTX A5000 GPUs. The code needed to reproduce the reported results is available online[2].

### A.1  Datasets

Table 7: Statistics of datasets used in the experiments.

| DATASETS | Type | Time steps | Nodes | Edges | Rate |
|----------|------|------------|-------|-------|------|
| GPVAR | Undirected | 30,000 | 120 | 199 | N/A |
| GPVAR-L | Undirected | 30,000 | 120 | 199 | N/A |
| METR-LA | Directed | 34,272 | 207 | 1515 | 5 minutes |
| PEMS-BAY | Directed | 52,128 | 325 | 2369 | 5 minutes |
| CER-E | Directed | 25,728 | 485 | 4365 | 30 minutes |
| AQI | Undirected | 8,760 | 437 | 2699 | 1 hour |
| PEMS03 | Directed | 26,208 | 358 | 546 | 5 minutes |
| PEMS04 | Directed | 16,992 | 307 | 340 | 5 minutes |
| PEMS07 | Directed | 28,224 | 883 | 866 | 5 minutes |
| PEMS08 | Directed | 17,856 | 170 | 277 | 5 minutes |

In this section, we provide additional information regarding each one of the considered datasets. Tab. 7 reports a summary of statistics.

**GPVAR(-L)**  For the GPVAR datasets we follow the procedure described in Sec. 7 to generate data and then partition the resulting time series in $70\%/10\%/20\%$ splits for training, validation and testing, respectively. For GPVAR-L the parameters of the spatiotemporal process are set as

$$\Theta = \begin{bmatrix} 2.5 & -2.0 & -0.5 \\ 1.0 & 3.0 & 0.0 \end{bmatrix}, \qquad \boldsymbol{a}, \boldsymbol{b} \sim \mathcal{U}\left(-2, 2\right),$$

$$\eta \sim \mathcal{N}(\mathbf{0}, \mathrm{diag}(\sigma^2)), \quad \sigma = 0.4.$$

Fig. 2 shows the topology of the graph used as a support to the process. In particular, we considered a network with 120 nodes with 20 communities and added self-loops to the graph adjacency matrix.

---

[2]https://github.com/Graph-Machine-Learning-Group/taming-local-effects-stgnns

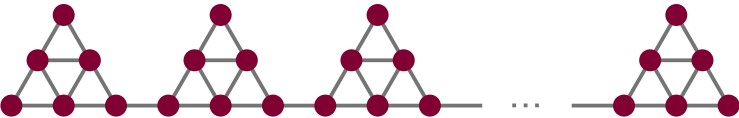

Figure 2: GPVAR community graph. We used a graph with 20 communities resulting in a network with 120 nodes.

**Spatiotemporal forecasting benchmarks**   We considered several datasets coming from relevant application domains and different problem settings corresponding to real-world application scenarios.

**Traffic forecasting**  We consider two popular traffic forecasting datasets, namely **METR-LA** and **PEMS-BAY** [6], containing measurements from loop detectors in the Los Angeles County Highway and San Francisco Bay Area, respectively. For the experiment on transfer learning, we use the **PEMS03**, **PEMS04**, **PEMS07**, and **PEMS08** datasets from Guo et al. [56] each collecting traffic flow readings, aggregated into 5-minutes intervals, from different areas in California provided by Caltrans Performance Measurement System (PeMS) [64].

**Electric load forecasting**  We selected the **CER-E** dataset [54], a collection of energy consumption readings, aggregated into 30-minutes intervals, from 485 smart meters monitoring small and medium-sized enterprises.

**Air quality monitoring**  The **AQI** [55] dataset collects hourly measurements of pollutant PM2.5 from 437 air quality monitoring stations in China. Note that all of these datasets have been previously used for spatiotemporal forecasting and imputation [47, 65].

For each dataset, we obtain the corresponding adjacency matrix by following previous works [20, 6, 56]. We use as exogenous variables sinusoidal functions encoding the time of the day and the one-hot encoding of the day of the week. For datasets with an excessive number of missing values, namely METR-LA and AQI, we add as an exogenous variable also a binary mask indicating if the corresponding value has been imputed.

We split datasets into windows of $W$ time steps, and train the models to predict the next $H$ observations. For the experiment in Tab.4, we set $W = 12, H = 12$ for the traffic datasets, $W = 48, H = 6$ for CER-E, and $W = 24, H = 3$ for AQI. Then, for all datasets except for AQI, we divide the obtained windows sequentially into $70\%/10\%/20\%$ splits for training, validation, and testing, respectively. For AQI instead, we use as the test set the months of March, June, September, and December, following Yi et al. [66].

## A.2   Architectures

In this section, we provide a detailed description of the reference architectures used in the study.

**Reference architectures**   We consider 4 simple STGNN architectures that follow the template given in (Eq. 9–11). In all the models, we use as ENCODER (Eq. 9) a simple linear layer s.t.

$$\boldsymbol{h}_t^{i,0} = \boldsymbol{W}_{\text{enc}}\big[\boldsymbol{x}_{t-1}^i || \boldsymbol{u}_{t-1}^i\big] \tag{21}$$

and as DECODER (Eq. 11) a 2 layer module structured as

$$\hat{\boldsymbol{x}}_{t:t+H}^i = \Big\{\boldsymbol{W}_h\xi\Big(\boldsymbol{W}_{\text{dec}}\boldsymbol{h}_t^{i,L}\Big)\Big\}_{h=1,\dots,H} \tag{22}$$

where $\boldsymbol{W}_{\text{enc}}$, $\boldsymbol{W}_{\text{dec}}$, and $\boldsymbol{W}_1,\dots,\boldsymbol{W}_H$ are learnable parameters. In hybrid models with local ENCODER and/or DECODER, we use different parameter matrices for each node, i.e.,

$$\boldsymbol{h}_t^{i,0} = \boldsymbol{W}_{\text{enc}}^i\big[\boldsymbol{x}_{t-1}^i || \boldsymbol{u}_{t-1}^i\big] \tag{23}$$

$$\hat{\boldsymbol{x}}_{t:t+H}^i = \Big\{\boldsymbol{W}_h^i\xi\Big(\boldsymbol{W}_{\text{dec}}\boldsymbol{h}_t^{i,L}\Big)\Big\}_{h=1,\dots,H}. \tag{24}$$

When instead node embeddings $\boldsymbol{V} \in \mathbb{R}^{N\times d_v}$ are used to specialize the model, they are concatenated to the modules' input as

$$\boldsymbol{h}_t^{i,0} = \boldsymbol{W}_{\text{enc}}\big[\boldsymbol{x}_{t-1}^i || \boldsymbol{u}_{t-1}^i || \boldsymbol{v}^i\big] \tag{25}$$

$$\hat{\boldsymbol{x}}_{t:t+H}^i = \Big\{\boldsymbol{W}_h\xi\Big(\boldsymbol{W}_{\text{dec}}\big[\boldsymbol{h}_t^{i,L} || \boldsymbol{v}^i\big]\Big)\Big\}_{h=1,\dots,H}. \tag{26}$$

For the TTS architectures, we define the STMP module (Eq. 10) as a node-wise GRU [18]:

$$r_t^i = \sigma\big(W_1^l[h_t^{i,0} \| h_{t-1}^{i,1}]\big), \tag{27}$$

$$o_t^i = \sigma\big(W_2^l[h_t^{i,0} \| h_{t-1}^{i,1}]\big), \tag{28}$$

$$c_t^i = \tanh\big(W_3^l[h_t^{i,0} \| r_t^i \odot h_{t-1}^{i,1}]\big), \tag{29}$$

$$h_t^{i,1} = o_t^i \odot h_{t-1}^{i,1} + (1 - o_t^i) \odot c_t^i. \tag{30}$$

The GRU is then followed by $L$ MP layers

$$H_t^{l+1} = \mathrm{MP}^l\left(H_t^l, A\right), \quad l = 1, \ldots, L-1. \tag{31}$$

We consider two variants for this architecture: **TTS-IMP**, featuring the isotropic MP operator defined in Eq. 5, and **TTS-AMP**, using the anisotropic MP operator defined in Eq. 6-7. Note that all the parameters in the STMP blocks are shared among the nodes.

In T&S models, instead, we use a GRU where gates are implemented using MP layers:

$$r_t^{i,l} = \sigma\left(\mathrm{MP}_r^l\left(\left[h_t^{i,l-1}\|h_{t-1}^{i,l}\right], A\right)\right), \tag{32}$$

$$o_t^{i,l} = \sigma\left(\mathrm{MP}_o^l\left(\left[h_t^{i,l-1}\|h_{t-1}^{i,l}\right], A\right)\right), \tag{33}$$

$$c_t^{i,l} = \tanh\left(\mathrm{MP}_c^l\left(\left[h_t^{i,l-1}\|r_t^{i,l} \odot h_{t-1}^{i,l}\right], A\right)\right), \tag{34}$$

$$h_t^{i,l} = o_t^{i,l} \odot h_{t-1}^{i,l} + (1 - o_t^{i,l}) \odot c_t^{i,l}. \tag{35}$$

Similarly to the TTS case, we indicate as **T&S-IMP** the reference architecture in which MP operators are isotropic and **T&S-AMP** the one featuring anisotropic MP.

**Baselines**  For the RNN baselines we follow the same template of the reference architectures (Eq. 9-11) but use a single GRU cell as core processing module instead of the STMP block. The global-local **RNN**, then, uses node embeddings at encoding as shown in Eq. 25, whereas **LocalRNNs** have different sets of parameters for each time series. The **FC-RNN** architecture, instead, takes as input the concatenation all the time series as if they were a single multivariate one.

**Hyperparameters**  For the reference TTS architectures we use a GRU with a single cell followed by 2 message-passing layers. In T&S case we use a single graph recurrent convolutional cell. The number of neurons in each layer is set to $64$ and the embedding size to $32$ for all the reference architectures in all the benchmark datasets. Analogous hyperparameters were used for the RNN baselines. For GPVAR instead, we use $16$ and $8$ as hidden and embedding sizes, respectively. For the baselines from the literature, we use the hyperparameters used in the original papers whenever possible.

For GPVAR experiments we use a batch size of $128$ and train with early stopping for a maximum of $200$ epochs with the Adam optimizer [67] and a learning rate of $0.01$ halved every $50$ epochs.

For experiments in Tab. 1 and 4, we instead trained the models with batch size $64$ for a maximum of $300$ epochs each consisting of maximum $300$ batches. The initial learning rate was set to $0.003$ and reduced every $50$ epochs.

**Transfer learning experiment**  In the transfer learning experiments we simulate the case where a pre-trained spatiotemporal model for traffic forecasting is tested on a new road network. We fine-tune on each of the PEMS03-08 datasets models previously trained on the left-out 3 datasets. Following previous works [56], we split the datasets into $60\%/20\%/20\%$ for training, validation, and testing, respectively. We obtain the mini-batches during training by uniformly sampling $64$ windows from the 3 training sets. To train the models we use a similar experimental setting of experiments in Tab. 4, decreasing the number of epochs to $150$ and increasing the batches per epoch to $500$. We then assume that 2 weeks of readings from the target dataset are available for fine-tuning, and use the first week for training and the second one as the validation set. Then, we test fine-tuned models on the immediately following week (Tab. 5). For the global model, we either tune all the parameters or none of them (zero-shot setting). For fine-tuning, we increase the maximum number of epochs to 2000 without limiting the batches processed per epoch and fixing the learning rate to $0.001$. At the end of every

Table 8: Perfomance (MAE) of TTS-IMP and TTS-AMP variants and number of associated trainable parameters in PEMS-BAY (5-run average).

| | | | METR-LA | PEMS-BAY | (# weights) | METR-LA | PEMS-BAY | (# weights) |
|---|---|---|---|---|---|---|---|---|
| | | | TTS-IMP | | | TTS-AMP | | |
| | | Global TTS | $3.35 \pm 0.01$ | $1.72 \pm 0.00$ | $4.71 \times 10^4$ | $3.27 \pm 0.01$ | $1.68 \pm 0.00$ | $6.41 \times 10^4$ |
| Global-local TTS | Weights | Encoder | $3.15 \pm 0.01$ | $1.66 \pm 0.01$ | $2.75 \times 10^5$ | $3.12 \pm 0.01$ | $1.64 \pm 0.00$ | $2.92 \times 10^5$ |
| | | Decoder | $3.09 \pm 0.01$ | $1.58 \pm 0.00$ | $3.00 \times 10^5$ | $3.09 \pm 0.01$ | $1.58 \pm 0.00$ | $3.17 \times 10^5$ |
| | | Enc. + Dec. | $3.16 \pm 0.01$ | $1.70 \pm 0.01$ | $5.28 \times 10^5$ | $3.14 \pm 0.01$ | $1.69 \pm 0.01$ | $5.45 \times 10^5$ |
| | Embed. | Encoder | $3.08 \pm 0.01$ | $1.58 \pm 0.00$ | $5.96 \times 10^4$ | $3.05 \pm 0.02$ | $1.58 \pm 0.01$ | $7.66 \times 10^4$ |
| | | Decoder | $3.13 \pm 0.00$ | $1.60 \pm 0.00$ | $5.96 \times 10^4$ | $3.12 \pm 0.01$ | $1.60 \pm 0.00$ | $7.66 \times 10^4$ |
| | | Enc. + Dec. | $3.07 \pm 0.01$ | $1.58 \pm 0.00$ | $6.16 \times 10^4$ | $3.04 \pm 0.01$ | $1.59 \pm 0.01$ | $7.86 \times 10^4$ |
| | | FC-RNN | $3.56 \pm 0.03$ | $2.32 \pm 0.01$ | $3.04 \times 10^5$ | | | |
| | | LocalRNNs | $3.69 \pm 0.00$ | $1.91 \pm 0.00$ | $1.10 \times 10^7$ | | | |

training epoch, we compute the MAE on the validation set and stop training if it has not decreased in the last 100 epochs, restoring the model weights corresponding to the best-performing model. For the global-local models with variational regularization on the embedding space, during training, we set $\beta = 0.05$ and initialize the distribution parameters as

$$\boldsymbol{\mu}_i \sim \mathcal{U}\left(-0.01, 0.01\right), \qquad \boldsymbol{\sigma}_i = \mathbf{0.2}.$$

For fine-tuning instead, we initialize the new embedding table $\boldsymbol{V}' \in \mathbb{R}^{N' \times d_v}$ as

$$\boldsymbol{V}' \sim \mathcal{U}\left(-\Delta, \Delta\right),$$

where $\Delta = \frac{1}{\sqrt{d_v}}$ and remove the regularization loss. For the clustering regularization, instead, we use $K = 10$ clusters and regularization trade-off weight $\lambda = 0.5$. We initialize embedding, centroid, and cluster assignment matrices as

$$\boldsymbol{V} \sim \mathcal{U}\left(-\Delta, \Delta\right) \quad \boldsymbol{C} \sim \mathcal{U}\left(-\Delta, \Delta\right) \quad \boldsymbol{S} \sim \mathcal{U}\left(0, 1\right)$$

respectively. For fine-tuning, we fix the centroid table $\boldsymbol{C}$ and initialize the new embedding table $\boldsymbol{V}' \in \mathbb{R}^{N' \times d_v}$ and cluster assignment matrix $\boldsymbol{S}' \in \mathbb{R}^{N' \times K}$ following an analogous procedure. Finally, we increase the regularization weight $\lambda$ to 10.

## B  Additional experimental results

**Local components**    Table 8, an extended version of Table 1, shows the performance of reference architecture with and without local components. Here we consider TTS-IMP and TTS-AMP, together with FC-RNN (a multivariate RNN) and LocalRNNs (local univariate RNNs with a different set of parameters for each time series). For the STGNNs, we consider a global variant (without any local component) and global-local alternatives, where we insert node-specific components within the architecture by means of (1) different sets of weights for each time series (light brown block) or (2) node embeddings (light gray block). More precisely, we show how performance varies when the local components are added in the encoding and decoding steps together, or uniquely in one of the two steps. Results for TTS-AMP are consistent with the observations made for TTS-IMP, showing that the use of local components enables improvements up to 7% w.r.t. the fully global variant.

**Transfer learning**    Tables 9 to 12 show additional results for the transfer learning experiments in all the target datasets. In particular, each table shows results for the reference architectures w.r.t. different training set sizes (from 1 day to 2 weeks) and considers the settings where embeddings are fed to both encoder and decoder or decoder only. We report results on the test data corresponding to the week after the validation set but also on the original test split used in the literature. In the last columns of the table, we also show the performance that one would have obtained 100 epochs after the minimum in the validation error curve; the purpose of showing these results is to hint at the performance that one would have obtained without holding out 1 week of data for validation. The results indeed suggest that fine-tuning the full global model is more prone to overfitting.

Table 9: Forecasting error (MAE) on PEMS03 in the transfer learning setting (5 runs average).

| Model | Testing on 1 subsequent week | | | | Testing on the standard split | | | | Testing 100 epochs after validation min. | | | |
|---|---|---|---|---|---|---|---|---|---|---|---|---|
| TTS-IMP | 2 weeks | 1 week | 3 days | 1 day | 2 weeks | 1 week | 3 days | 1 day | 2 weeks | 1 week | 3 days | 1 day |
| Global | $14.86_{\pm.02}$ | $15.30_{\pm.03}$ | $16.26_{\pm.08}$ | $16.65_{\pm.07}$ | $16.11_{\pm.05}$ | $16.36_{\pm.05}$ | $16.95_{\pm.04}$ | $17.39_{\pm.12}$ | $16.30_{\pm.10}$ | $16.58_{\pm.07}$ | $17.62_{\pm.23}$ | $18.33_{\pm.32}$ |
| ENC.+DEC. Embeddings | $14.53_{\pm.02}$ | $14.64_{\pm.05}$ | $15.87_{\pm.08}$ | $16.78_{\pm.12}$ | $16.03_{\pm.05}$ | $16.12_{\pm.05}$ | $17.18_{\pm.16}$ | $17.82_{\pm.15}$ | $16.09_{\pm.06}$ | $16.16_{\pm.05}$ | $17.28_{\pm.18}$ | $17.94_{\pm.17}$ |
| ENC.+DEC. – Variational | $14.50_{\pm.04}$ | $14.56_{\pm.03}$ | $15.40_{\pm.06}$ | $15.65_{\pm.11}$ | $15.69_{\pm.10}$ | $15.70_{\pm.12}$ | $16.33_{\pm.06}$ | $16.52_{\pm.10}$ | $15.70_{\pm.10}$ | $15.70_{\pm.12}$ | $16.35_{\pm.07}$ | $16.54_{\pm.10}$ |
| ENC.+DEC. – Clustering | $14.58_{\pm.02}$ | $14.60_{\pm.02}$ | $15.67_{\pm.08}$ | $16.53_{\pm.13}$ | $15.65_{\pm.07}$ | $15.71_{\pm.08}$ | $16.76_{\pm.15}$ | $17.76_{\pm.15}$ | $15.70_{\pm.05}$ | $15.74_{\pm.07}$ | $16.80_{\pm.14}$ | $17.78_{\pm.14}$ |
| DEC. Embeddings | $14.79_{\pm.02}$ | $14.84_{\pm.03}$ | $15.49_{\pm.03}$ | $16.01_{\pm.08}$ | $16.06_{\pm.05}$ | $16.12_{\pm.07}$ | $16.74_{\pm.04}$ | $17.25_{\pm.07}$ | $16.08_{\pm.05}$ | $16.13_{\pm.07}$ | $16.75_{\pm.04}$ | $17.29_{\pm.06}$ |
| DEC. – Variational | $15.33_{\pm.03}$ | $15.39_{\pm.02}$ | $15.83_{\pm.04}$ | $16.03_{\pm.04}$ | $16.15_{\pm.02}$ | $16.20_{\pm.02}$ | $16.60_{\pm.04}$ | $16.75_{\pm.06}$ | $16.15_{\pm.02}$ | $16.20_{\pm.02}$ | $16.60_{\pm.04}$ | $16.76_{\pm.06}$ |
| DEC. – Clustering | $14.96_{\pm.06}$ | $15.09_{\pm.06}$ | $15.88_{\pm.07}$ | $15.81_{\pm.03}$ | $16.25_{\pm.08}$ | $16.29_{\pm.06}$ | $16.72_{\pm.08}$ | $16.87_{\pm.07}$ | $16.28_{\pm.07}$ | $16.19_{\pm.05}$ | $16.91_{\pm.10}$ | $16.93_{\pm.07}$ |

Table 10: Forecasting error (MAE) on PEMS04 in the transfer learning setting (5 runs average).

| Model | Testing on 1 subsequent week. | | | | Testing on the standard split | | | | Testing 100 epochs after validation min. | | | |
|---|---|---|---|---|---|---|---|---|---|---|---|---|
| TTS-IMP | 2 weeks | 1 week | 3 days | 1 day | 2 weeks | 1 week | 3 days | 1 day | 2 weeks | 1 week | 3 days | 1 day |
| Global | $20.86_{\pm.03}$ | $21.59_{\pm.11}$ | $21.84_{\pm.06}$ | $22.26_{\pm.10}$ | $20.79_{\pm.02}$ | $21.68_{\pm.05}$ | $22.10_{\pm.10}$ | $22.59_{\pm.11}$ | $20.89_{\pm.05}$ | $21.97_{\pm.13}$ | $22.88_{\pm.07}$ | $23.85_{\pm.22}$ |
| ENC.+DEC. Embeddings | $19.96_{\pm.08}$ | $20.27_{\pm.11}$ | $21.03_{\pm.14}$ | $21.99_{\pm.13}$ | $19.87_{\pm.07}$ | $20.27_{\pm.05}$ | $21.20_{\pm.15}$ | $22.38_{\pm.14}$ | $19.88_{\pm.07}$ | $20.29_{\pm.07}$ | $21.28_{\pm.14}$ | $22.46_{\pm.14}$ |
| ENC.+DEC. – Variational | $19.94_{\pm.08}$ | $20.19_{\pm.05}$ | $20.71_{\pm.12}$ | $21.20_{\pm.15}$ | $19.92_{\pm.06}$ | $20.23_{\pm.05}$ | $20.82_{\pm.08}$ | $21.46_{\pm.13}$ | $19.92_{\pm.06}$ | $20.23_{\pm.05}$ | $20.82_{\pm.08}$ | $21.47_{\pm.12}$ |
| ENC.+DEC. – Clustering | $19.69_{\pm.06}$ | $19.91_{\pm.11}$ | $20.48_{\pm.09}$ | $21.91_{\pm.21}$ | $19.70_{\pm.06}$ | $19.96_{\pm.11}$ | $20.62_{\pm.09}$ | $22.28_{\pm.21}$ | $19.72_{\pm.07}$ | $19.97_{\pm.10}$ | $20.65_{\pm.08}$ | $22.29_{\pm.21}$ |
| DEC. Embeddings | $20.10_{\pm.06}$ | $20.27_{\pm.04}$ | $20.87_{\pm.08}$ | $21.44_{\pm.09}$ | $20.18_{\pm.07}$ | $20.39_{\pm.05}$ | $21.01_{\pm.09}$ | $21.70_{\pm.08}$ | $20.19_{\pm.07}$ | $20.40_{\pm.05}$ | $21.03_{\pm.08}$ | $21.74_{\pm.08}$ |
| DEC. – Variational | $20.79_{\pm.06}$ | $20.94_{\pm.05}$ | $21.23_{\pm.07}$ | $21.51_{\pm.06}$ | $20.94_{\pm.06}$ | $21.10_{\pm.05}$ | $21.40_{\pm.08}$ | $21.76_{\pm.05}$ | $20.94_{\pm.06}$ | $21.10_{\pm.05}$ | $21.40_{\pm.08}$ | $21.77_{\pm.05}$ |
| DEC. – Clustering | $20.19_{\pm.09}$ | $20.45_{\pm.10}$ | $20.63_{\pm.06}$ | $21.03_{\pm.05}$ | $20.27_{\pm.09}$ | $20.56_{\pm.10}$ | $20.81_{\pm.06}$ | $21.28_{\pm.06}$ | $20.75_{\pm.05}$ | $20.78_{\pm.05}$ | $20.89_{\pm.05}$ | $21.35_{\pm.05}$ |

Table 11: Forecasting error (MAE) on PEMS07 in the transfer learning setting (5 runs average).

| Model | Testing on 1 subsequent week | | | | Testing on the standard split | | | | Testing 100 epochs after validation min. | | | |
|---|---|---|---|---|---|---|---|---|---|---|---|---|
| TTS-IMP | 2 weeks | 1 week | 3 days | 1 day | 2 weeks | 1 week | 3 days | 1 day | 2 weeks | 1 week | 3 days | 1 day |
| Global | $22.87_{\pm.05}$ | $23.82_{\pm.03}$ | $24.52_{\pm.06}$ | $25.40_{\pm.06}$ | $22.64_{\pm.04}$ | $23.58_{\pm.02}$ | $24.20_{\pm.06}$ | $25.04_{\pm.06}$ | $22.72_{\pm.05}$ | $23.72_{\pm.02}$ | $24.45_{\pm.11}$ | $25.48_{\pm.05}$ |
| ENC.+DEC. Embeddings | $21.68_{\pm.07}$ | $22.23_{\pm.08}$ | $23.54_{\pm.19}$ | $26.11_{\pm.61}$ | $22.10_{\pm.09}$ | $22.77_{\pm.05}$ | $24.17_{\pm.24}$ | $26.79_{\pm.63}$ | $22.11_{\pm.09}$ | $22.78_{\pm.05}$ | $24.21_{\pm.22}$ | $26.82_{\pm.63}$ |
| ENC.+DEC. – Variational | $22.05_{\pm.05}$ | $22.43_{\pm.02}$ | $23.23_{\pm.08}$ | $24.40_{\pm.13}$ | $22.18_{\pm.04}$ | $22.59_{\pm.04}$ | $23.44_{\pm.11}$ | $24.62_{\pm.13}$ | $22.18_{\pm.04}$ | $22.59_{\pm.04}$ | $23.44_{\pm.10}$ | $24.62_{\pm.13}$ |
| ENC.+DEC. – Clustering | $21.75_{\pm.05}$ | $22.16_{\pm.07}$ | $23.36_{\pm.20}$ | $26.44_{\pm.26}$ | $22.03_{\pm.08}$ | $22.52_{\pm.10}$ | $23.85_{\pm.24}$ | $27.12_{\pm.27}$ | $22.03_{\pm.09}$ | $22.55_{\pm.11}$ | $23.85_{\pm.24}$ | $27.13_{\pm.28}$ |
| DEC. Embeddings | $22.50_{\pm.14}$ | $22.83_{\pm.13}$ | $23.59_{\pm.12}$ | $24.89_{\pm.19}$ | $22.68_{\pm.12}$ | $23.13_{\pm.10}$ | $24.04_{\pm.09}$ | $25.41_{\pm.20}$ | $22.69_{\pm.12}$ | $23.14_{\pm.10}$ | $24.04_{\pm.09}$ | $25.43_{\pm.20}$ |
| DEC. – Variational | $24.32_{\pm.16}$ | $24.60_{\pm.16}$ | $25.12_{\pm.17}$ | $25.50_{\pm.15}$ | $24.25_{\pm.14}$ | $24.60_{\pm.13}$ | $25.16_{\pm.13}$ | $25.56_{\pm.12}$ | $24.25_{\pm.14}$ | $24.60_{\pm.13}$ | $25.16_{\pm.13}$ | $25.57_{\pm.12}$ |
| DEC. – Clustering | $23.02_{\pm.09}$ | $23.53_{\pm.09}$ | $24.42_{\pm.15}$ | $24.87_{\pm.13}$ | $23.18_{\pm.09}$ | $23.77_{\pm.08}$ | $24.66_{\pm.12}$ | $25.24_{\pm.09}$ | $23.91_{\pm.16}$ | $24.10_{\pm.15}$ | $24.73_{\pm.10}$ | $25.27_{\pm.09}$ |

Table 12: Forecasting error (MAE) on PEMS08 in the transfer learning setting (5 runs average).

| Model | Testing on 1 subsequent week | | | | Testing on the standard split | | | | Testing 100 epochs after validation min. | | | |
|---|---|---|---|---|---|---|---|---|---|---|---|---|
| TTS-IMP | 2 weeks | 1 week | 3 days | 1 day | 2 weeks | 1 week | 3 days | 1 day | 2 weeks | 1 week | 3 days | 1 day |
| Global | $15.51_{\pm.03}$ | $15.90_{\pm.07}$ | $16.87_{\pm.05}$ | $17.59_{\pm.04}$ | $15.36_{\pm.03}$ | $15.71_{\pm.06}$ | $16.71_{\pm.06}$ | $17.41_{\pm.03}$ | $15.47_{\pm.06}$ | $15.87_{\pm.04}$ | $17.58_{\pm.16}$ | $18.46_{\pm.12}$ |
| ENC.+DEC. Embeddings | $15.45_{\pm.08}$ | $15.45_{\pm.06}$ | $16.34_{\pm.07}$ | $17.15_{\pm.08}$ | $15.34_{\pm.07}$ | $15.32_{\pm.04}$ | $16.27_{\pm.07}$ | $17.11_{\pm.08}$ | $15.32_{\pm.09}$ | $15.30_{\pm.03}$ | $16.27_{\pm.05}$ | $17.13_{\pm.08}$ |
| ENC.+DEC. – Variational | $15.34_{\pm.04}$ | $15.41_{\pm.06}$ | $15.83_{\pm.07}$ | $16.32_{\pm.11}$ | $15.21_{\pm.03}$ | $15.27_{\pm.06}$ | $15.70_{\pm.06}$ | $16.19_{\pm.12}$ | $15.21_{\pm.03}$ | $15.27_{\pm.05}$ | $15.69_{\pm.06}$ | $16.19_{\pm.12}$ |
| ENC.+DEC. – Clustering | $15.41_{\pm.06}$ | $15.41_{\pm.06}$ | $15.96_{\pm.04}$ | $16.99_{\pm.07}$ | $15.27_{\pm.07}$ | $15.27_{\pm.07}$ | $15.90_{\pm.05}$ | $16.99_{\pm.08}$ | $15.30_{\pm.08}$ | $15.28_{\pm.07}$ | $15.90_{\pm.05}$ | $17.02_{\pm.10}$ |
| DEC. Embeddings | $15.72_{\pm.06}$ | $15.74_{\pm.06}$ | $16.41_{\pm.07}$ | $16.97_{\pm.08}$ | $15.61_{\pm.06}$ | $15.61_{\pm.06}$ | $16.33_{\pm.08}$ | $16.90_{\pm.09}$ | $15.61_{\pm.06}$ | $15.61_{\pm.06}$ | $16.35_{\pm.08}$ | $16.92_{\pm.10}$ |
| DEC. – Variational | $16.31_{\pm.10}$ | $16.33_{\pm.15}$ | $16.53_{\pm.15}$ | $16.74_{\pm.12}$ | $16.13_{\pm.09}$ | $16.14_{\pm.14}$ | $16.34_{\pm.13}$ | $16.55_{\pm.11}$ | $16.13_{\pm.09}$ | $16.14_{\pm.13}$ | $16.35_{\pm.14}$ | $16.56_{\pm.11}$ |
| DEC. – Clustering | $15.81_{\pm.11}$ | $15.92_{\pm.14}$ | $16.11_{\pm.08}$ | $16.55_{\pm.10}$ | $15.70_{\pm.10}$ | $15.77_{\pm.11}$ | $15.97_{\pm.08}$ | $16.43_{\pm.10}$ | $15.98_{\pm.06}$ | $15.90_{\pm.06}$ | $16.01_{\pm.08}$ | $16.45_{\pm.11}$ |

