# OpenReview forum: "Taming Local Effects in Graph-based Spatiotemporal Forecasting"
_NeurIPS.cc/2023/Conference — NeurIPS 2023 poster_

### Official Review · Reviewer_Lyxe · 2023-06-12

**Soundness:** 3 good
**Presentation:** 3 good
**Contribution:** 3 good
**Rating:** 6
**Confidence:** 3

**Summary:**

This paper proposes a method to leverage local effects in graph-based spatio-temporal forecasting. The authors claim that existing spatio-temporal graph neural networks are global models, i.e. all nodes share the same set of parameters, and thus may fail to capture some node-specific patterns. On the other hand, local models, in which some layers within the models are node-specifically parameterized, have better performances compared to global ones, but at the cost of many additional parameters. The authors find a method, random node embeddings, to strike a balance between local and global methods. The authors also propose regularizations to improve transferability of the node embeddings and the resulting models. Experiments over real-world data are given where the proposed method achieves consistent improvements over a variety of models and datasets.

**Strengths:**

1. The studied problem is interesting. It challenges the assumption that a shared global STGNN is used for all nodes, which is standard in previous works.
2. The proposed technique, i.e. trainable node embeddings, is simple and sound. The regularization terms designed also make sense.
3. The proposed technique with trainable node embeddings are effective over various models (DCRNN, AGCRN, GWNet) and real-world datasets, which shows the generality of the proposed technique.
4. The experimental results showing that node embeddings and regularizations are effective in terms of knowledge transfer are a plus. Intuitively, people may think that models with node specific parameters will not perform well in unseen nodes, but the results show the opposite.

**Weaknesses:**

1. The proposed method with node specific embeddings is effective, but not new. Specifically, STID [40] proposes exactly the same technique in terms of trainable node embeddings. I am slightly concerned about whether the technical contribution meets the standard of NeurIPS with this existing work.
2. The fine-grained categorization about spatio-temporal graph neural networks do not seem necessary, e.g. T&S, TTS and anistropic VS istropic. I fail to see how introducing these concepts help better understand the paper, and thus I would suggest these parts be removed.

**Questions:**

I really like this paper and since it is in general clearly written, I do not have questions at this time.

---

> ### Author Rebuttal · Authors · 2023-08-08
>
> Thank you for your comments and your positive opinion about our work. Please find our point-by-point answers below.
>
> > 1. The proposed method with node specific embeddings is effective, but not new. Specifically, STID [40] proposes exactly the same technique in terms of trainable node embeddings. I am slightly concerned about whether the technical contribution meets the standard of NeurIPS with this existing work.
>
> Similar node embeddings have been used in some architectures. In this paper, we rationalize the practice of introducing such trainable components by providing an explanatory framework for the observed empirical results. We believe our contribution fully meets NeurIPS standards as it sheds light on extremely relevant challenges of very popular architectures. In fact, we provide a comprehensive methodology accounting for local components in several settings and across different (global) models. Our framework allows the practitioner to take full advantage of such hybrid models in both transductive and transfer learning settings.
>
> > 2. The fine-grained categorization about spatio-temporal graph neural networks do not seem necessary, e.g. T&S, TTS and anistropic VS istropic. I fail to see how introducing these concepts help better understand the paper, and thus I would suggest these parts be removed.
>
> The introduction of the different model architectures and design choices is necessary to show that the issues related to global and local aspects are present across a variety of architectures. In other words, the introduced categorization of existing architectures was necessary to carry out a proper and comprehensive empirical evaluation of the phenomena studied in the paper.

---

> > ### Comment · Reviewer_Lyxe · 2023-08-13
> > **Rebuttal acknowledged.**
> >
> > Thanks for your rebuttal.
> >
> > I like this paper and at this point I have no outstanding questions.

---

> > > ### Author Response · Authors · 2023-08-17
> > >
> > > Thank you for the feedback and the review!

---

### Official Review · Reviewer_jVAx · 2023-07-03

**Soundness:** 3 good
**Presentation:** 1 poor
**Contribution:** 3 good
**Rating:** 5
**Confidence:** 4

**Summary:**

This paper presents a methodological framework aimed at rationalizing the inclusion of trainable node embeddings in STGNNs for spatiotemporal forecasting applications. The authors examine the interplay between globality and locality in graph-based spatiotemporal forecasting and provide insights and guidelines for specification design. The paper demonstrates how incorporating trainable node embeddings in STGNNs can effectively combine the advantages of shared message-passing layers with node-specific parameters, while efficiently transferring the learned model to new node sets. The proposed framework is supported by empirical evidence and offers a principled approach for accommodating various node embeddings.

**Strengths:**

1. The authors investigate the interplay between globality and locality in graph-based spatiotemporal forecasting, resulting in five major findings.
2. The paper illustrates how including trainable node embeddings in STGNNs can effectively combine the benefits of shared message-passing layers with node-specific parameters and efficiently transfer the learned model to new node sets.


**Weaknesses:**

1. The paper is not well-organized, making it difficult to understand the main points and arguments presented.
2. The proposed framework adopts the TTS model as an STGNN, but some important TTS methods are not discussed in the related work, such as [1] and [2].
[1] Jianfei Gao and Bruno Ribeiro. On the equivalence between temporal and static equivariant graph representations. In International Conference on Machine Learning, pages 7052–7076. PMLR, 2022.
[2] Da Xu, etc. Inductive representation learning on temporal graphs. In ICLR 2020.


**Questions:**

The instructions in line 43 and lines 179-180 may appear contradictory. Could you provide a more detailed explanation?

**Limitations:**

see weakness

---

> ### Author Rebuttal · Authors · 2023-08-08
>
> Thanks for the review. Please find our answers below.
>
> > The paper is not well-organized, making it difficult to understand the main points and arguments presented.
>
> We did our best to make the structure of the paper easy to follow, an aspect that was appreciated by the other reviewers. Currently, the presentation is structured as follows: 1) template global architectures; 2) issues with fully global models; 3) introduction of hybrid global-local architectures (with empirical evidence of the benefits); 4) node embeddings (i.e., more efficient hybrid architectures); 5) overcoming the limitations of hybrid models in transfer learning settings; 6) complete empirical results. There’s always room for improvement and we are open to making adjustments based on more specific feedback that the reviewer wishes to provide.
>
> > The proposed framework adopts the TTS model as an STGNN, but some important TTS methods are not discussed in the related work, such as [1] and [2].
>
> We indicated that the terminology was indeed adapted from [1], we will add a reference to [2] in the related works. However, it should be noted that both [1,2] focus on temporal graphs rather than on time series. Finally, note that although we use TTS models as one of the reference architectures, the focus of the paper is not on introducing a new architecture but rather to study the impact of local effects on existing ones.
>
> > The instructions in line 43 and lines 179-180 may appear contradictory. Could you provide a more detailed explanation?
>
> The contradiction is only apparent. The point is that global models are indeed (in general) more efficient than local approaches [line 43] as the total number of parameters to be learned is smaller. However, if local effects are present in the data-generating process, this advantage might be compromised [lines 179-180], as more parameters might be needed to properly model them. This trade-off is what motivates the introduction of the hybrid models in the paper. We will make this apparent contradiction explicit in the paper. Thank you for the comment.

---

> > ### Comment · Reviewer_jVAx · 2023-08-16
> > **Rebuttal acknowledged**
> >
> > Thanks for your rebuttal and I would like to keep my score.

---

> > > ### Author Response · Authors · 2023-08-17
> > >
> > > Thank you for the answer.
> > >
> > > As far as we understood the main issue preventing a higher score was the organization of the paper, do you have any specific feedback about what is still unclear and on how we could improve? Have we satisfactorily addressed weakness 2 and the apparent contradiction in lines 43/179-180?

---

### Official Review · Reviewer_Zmtu · 2023-07-06

**Soundness:** 4 excellent
**Presentation:** 3 good
**Contribution:** 4 excellent
**Rating:** 7
**Confidence:** 3

**Summary:**

This paper examines the interaction between global and local effects in graph-based spatiotemporal forecasting. It addresses the limitations of a single global model by introducing a framework that incorporates trainable node embeddings into graph-based architectures. This framework enables the learning of specialized components and combines the benefits of shared message-passing layers with node-specific parameters. Additionally, the framework facilitates model transfer to new node sets. The paper offers empirical evidence and provides guidelines for adapting graph-based models to the dynamics of each time series to improve prediction accuracy.

**Strengths:**

It is nice to see a paper that investigates the attribution of "local" and "global" learning in modeling spatial-temporal graphs. The evaluation is very comprehensive and the paper is very informative. It may have great impact that can benefit the broad community that researches on spatial-temporal graphs.

**Weaknesses:**

The paper tries to answer a set of very big questions ("local" vs "global"), which I feel could be too hard to find a concrete answer in a 9 pages conference paper.

Similar questions can be asked for GNNs as well: Is message-passing more important or the node feature encoding more important? Should I go fully inductive like GCN, GraphSAGE? Or I just stick to non-inductive GNNs? Is the isotropic message-passing enough like vanilla GCNs, or do I need anisotropic message-passing like graph attention networks (GAT)? Do I interleave the MP layers with node encoding layers like most GCNs do? Or I should stack multiple node encoding layers before doing message-passing?

I feel it is a little too overwhelming to answer all these questions at once. I really appreciate the authors' efforts to investigate these questions, but it feel less convincing when it fits into a 9-pages conference paper, that each claim will be supported by less empirical evidences. Sometimes I will question that, how does this claim hold for other applications when the nature of a problem changes, would the conclusions change?

What's more challenging is that, these questions seem to not having a general answer that hold for all the applications, making it especially hard to draw conclusions by merely relying on empirical studies (or you will need a lot of experiments across much more domains).

In general, it is overall a technical solid paper and an ambitious one as well.

**Questions:**

I have to admit that I do not fully understand the paper, though I tried to read through the paper multiple times, so it may be helpful for me to understand the key contributions of the paper if the authors can provide some information of the following:

1. **What is the guideline and main takeaway messages for the audience to design models for spatial-temporal graph learning?** I feel a little lost in a vast amount of information and empirical observations.
2. **How do we know which design is the best for an application? Isn't it application specific?** If we want to know whether the "local" or "global" components in a STGNN are more important for an application, we may want to try it out or are there any ways to know it beforehand? To the best of my understanding, it could be very different from applications to applications, since global information is more important for some of them while for others local information is more important.
3. **Could T&S-AMP design be a generic to go choice?** If we do not know the importance of global or local effect in an application beforehand, can T&S and AMP be an to-go option? Many well-established GNNs for dynamic graphs fall in this category, for example, use attention in MP (AMP) and use alternating message-passing layers and recurrent layers (T&S), e.g., Graph Recurrent Attention Networks (GRAN).

**Limitations:**

No limitations and negative societal impacts are left unaddressed.

---

> ### Author Rebuttal · Authors · 2023-08-08
>
> Thank you for the detailed review and useful comments. We are happy that you found our paper interesting, please find our point-by-point answers below.
>
> > I feel it is a little too overwhelming to answer all these questions at once. [...] Sometimes I will question that, how does this claim hold for other applications when the nature of a problem changes, would the conclusions change?What's more challenging is that, these questions seem to not having a general answer.
>
> The paper tackles complex problems from different perspectives. In particular, we understand that the presentation of many different template architectures for STGNNs without providing guidelines on which architecture should be preferred in general might generate some confusion. However, we believe that introducing the many possible design choices is instrumental in showing that the issues related to global and local aspects (which are the focus of the paper) appear across the full spectrum of architectures used in practice. In such regard, we believe that the paper succeeds (through experiments on both synthetic and real-world datasets) in showing that locality and globality are crucial aspects in graph-based forecasting and in showing how local components can be effectively and efficiently introduced in otherwise global architectures.
>
> > Q1 What is the guideline and main takeaway messages for the audience to design models for spatial-temporal graph learning?
>
> The main takeaway messages are summarized in the 5 points in lines 64-75 in the paper.  In particular, 1) local components can be crucial to obtain accurate predictions in spatiotemporal forecasting; 2) node embeddings can amortize the learning of such components in otherwise global architectures; 3) hybrid local-global STGNNs with node embeddings can capture local effects with contained model capacity and reasonably long input window; 4) node embeddings make adapting models to different scenarios more efficient; 5) structuring the embedding space allow for regularizing the forecasting model.
>
> > Q2 How do we know which design is the best for an application? Isn't it application specific? If we want to know whether the "local" or "global" components in a STGNN are more important for an application, we may want to try it out or are there any ways to know it beforehand?
>
> Indeed it is application-specific. While fully global models are more flexible, hybrid architectures often perform better in practice. As there is no definitive answer, we suggest trying both architectures to decide whether including the local components is worth the compromise in flexibility. That being said, from our experience in real-world applications, adding local components consistently leads to better performance.
>
> > Q3 If we do not know the importance of global or local effect in an application beforehand, can T&S and AMP be an to-go option?
>
> A global-local T&S-AMP model is indeed a solid choice if the final performance at task is the only concern. However, T&S-AMP models are more computationally demanding than TTS-ISO models, which can nonetheless provide good performance. The practitioner should decide how to balance performance at task and computational costs and should be aware of the impact on the final performance of components that take local effects into account. The latter aspect is, as already mentioned, one of the main takeaways of the paper.

---

> > ### Comment · Reviewer_Zmtu · 2023-08-15
> >
> > I thank the authors for addressing my questions about the paper. Since my score has already acknowledged the contributions of this work, I will keep it as it is.

---

> > > ### Author Response · Authors · 2023-08-17
> > >
> > > Thank you for the feedback and thank you again for the review.

---

### Official Review · Reviewer_K1NT · 2023-07-07

**Soundness:** 3 good
**Presentation:** 2 fair
**Contribution:** 1 poor
**Rating:** 4
**Confidence:** 5

**Summary:**

In this paper, the authors explore the influence of locality and globality in graph-based spatiotemporal forecasting architectures. Existing spatiotemporal models are global trained on multiple multivariate timeseires, which can capture the strong dependency among individual nodes in a network. Standard local models such as RNNs learn each timeseries independently which lost the interaction information with other nodes, but are fitted solely on each individual trajectory resulting in good short-term prediction performance.

Directly combine the predictions from global and local models would result in a large number of model parameters (introduced by individual models). The authors instead propose to use a learnable embedding vector to represent the locality for each node and incorporate it in the GNN message passing procedure. The guide the learning process for such node embeddings, the authors further proposed two regularization terms to make the model more generalizable, with the assumption that the underlying dynamics of nodes within the same network topology would not differ too much.

Experiment result over several benchmark datasets show the proposed method is able to make better prediction results than compared baselines.



**Strengths:**

1. The writing for this paper is very easy to follow
2. The idea to inject local information into existing global spatiotemporal models is interesting.
3. The experiments are comprehensive, though some baselines are missing.

**Weaknesses:**

1. My major concern is the contribution/novelty of this paper. The authors propose to learn a node embedding to mimic the role of local models such as RNN trained on each individual timeseries. First of all, the node embeddings are static, whereas the output of RNN models are dynamic. Those local features for each individual node can changes over time which can be well-captured by any local models. Secondly, the learnable node embeddings seem to me are similar to those exogenous factors specific to each node, how to guarantee the learned embeddings would not serve as the same role as those exogenous factors? Finally, learning these embeddings would make the whole model not able to perform inductive tasks. When a new node/timeseries comes in, one needs to retrain the model instead of directly use the model to do the inference, opposed to existing spatiotemporal GNNs.

2. Also there are some missing baselines in terms of spatiotemporal GNNs, such as continuous graphODE approaches [1][2] and other discrete methods [3].

[1] Huang, Zijie, Yizhou Sun, and Wei Wang. "Learning continuous system dynamics from irregularly-sampled partial observations." Advances in Neural Information Processing Systems 33 (2020): 16177-16187.

[2] Song Wen, Hao Wang, and Dimitris Metaxas. 2022. Social ODE: Multi-agent Trajectory Forecasting with Neural Ordinary Differential Equations. In Computer Vision–ECCV 2022: 17th European Conference.

[3]Sanchez-Gonzalez, Alvaro, et al. "Learning to simulate complex physics with graph networks." International conference on machine learning. PMLR, 2020.

**Questions:**

1. Can the authors provide running time comparison during the testing stage, as the proposed method would need to retrain the model on unseen(new) nodes.
2. Can the authors visualize some of the learned local node embeddings and show some case study to interpret their semantic meanings?

**Limitations:**

The authors have not discuss the limitations of their model.

---

> ### Author Rebuttal · Authors · 2023-08-08
>
> Thank you for the review. Before providing point-by-point answers, we’d like to remark that the main contribution of our paper is not in introducing an architecture but rather in studying a crucial aspect of graph-based forecasting, i.e., the interplay of local and global aspects of time series forecasting in such architectures. We would appreciate it if the reviewer could reconsider their evaluation of the novelty/contributions of our paper in light of this.
>
> > The authors propose to learn a node embedding to mimic the role of local models such as RNN trained on each individual timeseries. First of all, the node embeddings are static, whereas the output of RNN models are dynamic.
>
> There might be some misunderstandings here. Node embeddings are learnable parameters and, as such, static once trained; the same holds true for the learnable parameters of an RNN, which are static as well; what is dynamic, instead, is the output of the models. Having node embeddings, rather than a fully local model, would only imply that most of the parameters involved in the dynamic processing of the data will be shared among time series. In other words, assuming the encoder of a global-local model is an RNN, the embeddings are passed as an additional input to provide localization, while the RNN parameters remain shared for all time series.
>
> > The learnable node embeddings seem to me are similar to those exogenous factors specific to each node, how to guarantee the learned embeddings would not serve as the same role as those exogenous factors?
>
> Embeddings are indeed used similarly to exogenous features and indeed exogenous features can be used to localize predictions. However, as specified in the paper, such features are often not available in practice, and node embeddings are far more flexible as the encoding is learned end-to-end, thus becoming part of the model’s parameters. Furthermore, structuring the embedding space allows for regularizing the local components of the model (as shown in Section 5.1 and the transfer learning experiments).
>
> > Learning these embeddings would make the whole model not able to perform inductive tasks. When a new node/timeseries comes in, one needs to retrain the model instead of directly use the model to do the inference, opposed to existing spatiotemporal GNNs.
>
> Yes, that is correct, adding learnable node embeddings makes the model not inductive. In this respect, a significant paper contribution is in showing that – with the proper regularizations – hybrid global-local models based on node embeddings can be adapted to new nodes using only a few observations, without training from scratch the full model. Finally, note that most of the state-of-the-art STGNN architectures are actually not inductive as they rely on some form of node identification (see, e.g., Graph Wavenet, AGCRN, etc.). We show that with our simple approach, we can get similar performance and that the resulting model can be easily transferred by fine-tuning only a very small number of parameters, drastically reducing sample complexity.
>
> > Also there are some missing baselines in terms of spatiotemporal GNNs, such as continuous graphODE approaches [1][2] and other discrete methods [3].
>
> In the empirical section, we focused on SOTA architectures for the benchmarks and types of problems considered in the study. However, even if the baselines suggested by the reviewer have been developed in a different context, we think it is indeed worth discussing them in the related works section. We will do so in the revision of the paper, thanks for the suggestion.
>
> > Q1 Can the authors provide running time comparison during the testing stage, as the proposed method would need to retrain the model on unseen(new) nodes?
>
> In the transductive setting, the models have the same computational costs as embedding results only in a small increase in the number of features and, as such, their impact w.r.t. time complexity is negligible.
> In the transfer learning setting, the computational cost is again exactly the same at inference time. The only overhead is the cost of fine-tuning the model (no full re-training is needed), yet such cost does not depend at all on the methodology we propose, but on the complexity of the model being fine-tuned and on the number of observations available. Furthermore, fine-tuning the entire model – rather than the embeddings alone – can be more computationally expensive. Finally, note that fine-tuning needs to be performed only once and that the performance improvement w.r.t. the zero-shot model is very large, even for inductive models.
>
> > Q2 Can the authors visualize some of the learned local node embeddings and show some case study to interpret their semantic meanings?
>
> Fig. 1 in the paper provides a visualization of the time series associated with different clusters of embeddings and, commenting on the figure, in Sec. 7 we discuss how the emerging clusters elucidate the role of embeddings as local components in the forecasting architectures. In addition to that, we include a t-SNE visualization of the learned embeddings with the different regularization mechanisms and different settings in the pdf attached to the rebuttal. The results confirm how regularization allows for structure to emerge in embedding space.
>
> > The authors have not discuss the limitations of their model.
>
> Limitations are discussed throughout the paper. In particular, we highlight several times the limits of the hybrid global-local model in the inductive settings and provide an in-depth discussion on the issue in Section 5.1. We will improve the discussion on the limitations of our study and include a comment on possible future works in the conclusions of the paper.

---

> > ### Comment · Reviewer_K1NT · 2023-08-15
> >
> > Thanks for your detailed response. Most of my concerns have been addressed so I raised my score to 4. But I still have questions regarding learning node embeddings for each timeseries and inject it in a shared global model for making predictions. As the authors mentioned that the node embeddings is similar to those exogenous factors, but the latter one are usually latent. Then is the proposed learnable node embeddings can be interpreted as latent exogenous factors?  It would be great if the authors can further summarize the differences between the two concepts.

---

> > > ### Author Response · Authors · 2023-08-17
> > > **Additional clarification**
> > >
> > > Sorry for the confusion, here’s a detailed discussion regarding the difference between the two.
> > >
> > > In our framework, node embeddings are in fact node-specific learnable parameters of the model (a different vector for each node) trained end-to-end together with the other (shared) model parameters. Using these node-specific trainable vectors allows us for tailoring (localize, in the terminology of the paper) the model predictions w.r.t. each time series. A global model (i.e., a model with no parameter specific to any time series) would not explicitly account for possible node-specific characteristics (local effects). Implementation-wise, once trained, node embeddings are passed as further inputs to the model, similar to how exogenous variables are typically processed.
> > >
> > > Exogenous variables, however, are usually additional covariates alongside the target time series. As an example, an exogenous variable can encode the day of the week or the external temperature. Although exogenous variables can be processed as additional inputs to forecasting models for conditioning the predictions, similarly to node embeddings, these are external inputs provided to the predictor and not learnable parameters associated with a specific node. The difference between the two is, then, quite large; what we meant in our previous answer is that node embeddings are used similarly to exogenous variables as they provide conditioning on the predictions.
> > >
> > > Indeed, as the reviewer suggests, the learned embeddings can be interpreted as latent factors conditioning the predictions. This interpretation motivates the regularizations proposed in Sec. 5.1. However, such latent vectors are learned directly, by parametrizing them with a separate set of learnable parameters for each time series. Also, once trained, these latent vectors are static: they are not conditioned on the current input window.
> > >
> > > We are available for providing further clarifications if needed, thanks again for the feedback and the careful review. We hope this addresses the issues currently preventing the reviewer from recommending acceptance.

---

### Author Rebuttal · Authors · 2023-08-08

We thank the reviewer for their insightful comments.

We provide point-by-point answers to each reviewer and attach as supplementary results a visualization of the embedding space for different regularization strategies for load and traffic forecasting datasets (see the attached pdf for more details).

We hope that the rebuttal clarifies all the raised issues.

---

### Decision · Program_Chairs · 2023-09-21

**Decision:**

Accept (poster)

**Comment:**

The paper proposes a novel graph-based spatiotemporal forecasting model that accounts for local effects. The reviewers think the notion of locality and embedding vector is novel and the experimental results are comprehensive. Please include the feedback and references provided by the reviewers in the final verison.